# The origins and consequences of *UPF1* variants in pancreatic adenosquamous carcinoma

Jacob T Polaski[1,2], Dylan B Udy[1,2,3], Luisa F Escobar-Hoyos[4,5,6,7], Gokce Askan[4], Steven D Leach[4,5,8,9], Andrea Ventura[10], Ram Kannan[10]*, Robert K Bradley[1,2]*

[1]Computational Biology Program, Public Health Sciences Division, Fred Hutchinson Cancer Research Center, Seattle, United States; [2]Basic Sciences Division, Fred Hutchinson Cancer Research Center, Seattle, United States; [3]Molecular and Cellular Biology Graduate Program, University of Washington, Seattle, United States; [4]David M. Rubenstein Center for Pancreatic Cancer Research, Memorial Sloan Kettering Cancer Center, New York, United States; [5]Human Oncology and Pathogenesis Program, Memorial Sloan Kettering Cancer Center, New York, United States; [6]Department of Pathology, Stony Brook University, New York, United States; [7]Yale University School of Medicine, New Haven, United States; [8]Department of Surgery, Memorial Sloan Kettering Cancer Center, New York, United States; [9]Dartmouth Norris Cotton Cancer Center, Lebanon, United States; [10]Cancer Biology and Genetics Program, Memorial Sloan Kettering Cancer Center, New York, United States

**Abstract** Pancreatic adenosquamous carcinoma (PASC) is an aggressive cancer whose mutational origins are poorly understood. An early study reported high-frequency somatic mutations affecting UPF1, a nonsense-mediated mRNA decay (NMD) factor, in PASC, but subsequent studies did not observe these lesions. The corresponding controversy about whether *UPF1* mutations are important contributors to PASC has been exacerbated by a paucity of functional studies. Here, we modeled two *UPF1* mutations in human and mouse cells to find no significant effects on pancreatic cancer growth, acquisition of adenosquamous features, *UPF1* splicing, UPF1 protein, or NMD efficiency. We subsequently discovered that 45% of *UPF1* mutations reportedly present in PASCs are identical to standing genetic variants in the human population, suggesting that they may be non-pathogenic inherited variants rather than pathogenic mutations. Our data suggest that *UPF1* is not a common functional driver of PASC and motivate further attempts to understand the genetic origins of these malignancies.

*For correspondence:
ramk.1019@gmail.com (RK);
rbradley@fredhutch.org (RKB)

**Competing interests:** The authors declare that no competing interests exist.

## Introduction

Pancreatic adenosquamous carcinoma (PASC) is a rare and aggressive disease that constitutes 1–4% of pancreatic exocrine tumors (*Madura et al., 1999*). Patient prognosis is extremely poor, with a median survival of 8 months (*Simone et al., 2013*). Although PASC is clinically and histologically distinct from the more common disease pancreatic adenocarcinoma, the genetic and molecular origins of PASC's unique features are unknown.

A recent study reported a potential breakthrough in our understanding of PASC etiology. *Liu et al., 2014* reported high-frequency mutations affecting *UPF1*, which encodes a core component of the nonsense-mediated mRNA decay (NMD) pathway, in 78% (18 of 23) of PASC patients. These mutations were absent from patient-matched normal pancreatic tissue (0 of 18) and from non-

**eLife digest** Cancer is a group of complex diseases in which cells grow uncontrollably and spread into surrounding tissues and other parts of the body. All types of cancers develop from changes – or mutations – in the genes that affect the pathways involved in controlling the growth of cells.

Different cancers possess unique sets of mutations that affect specific genes, and often, it is difficult to determine which of them play the most important role in a particular type of cancer. For example, pancreatic adenosquamous carcinoma, a rare and aggressive form of pancreatic cancer, is a devastating disease with a poor chance of survival – patients rarely live longer than one year after diagnosis.

While the cells of this particular cancer display distinct features that separate them from other forms of pancreatic cancer, the genetic causes of these features are unclear. Using new technologies, some researchers have reported mutations in a 'quality control' gene called 'UPF1', which is responsible for destroying faulty forms of genetic material. However, subsequent studies did not find such mutations.

To clarify the role of UPF1 in pancreatic adenosquamous carcinoma, Polaski et al. used mouse and human cancer cells with UPF1 mutations and monitored their effects on tumour growth and the development of features unique to this disease.

Polaski et al. first injected mice with mouse pancreatic cancer cells containing mutations in UPF1 (mutated cells) and cancer cells without. Both groups of mice developed pancreatic tumours but there was no difference in tumour growth between the mutated and non-mutated cells, and neither cell type displayed distinct features. The researchers then generated human mutated cells, which were also found to lack any specific characteristics. Further analysis showed that the mutations did not stop UPF1 from working, in fact, over 40% of these mutations occurred naturally in humans without causing cancer.

This suggests that UPF1 does not seem to be involved in pancreatic adenosquamous carcinoma. Further investigation is needed to illuminate key genetic players in the development of this type of cancer, which will be vital for improving treatments and outcomes for patients suffering from this disease.

PASC tumors (0 of 29 non-adenosquamous pancreatic carcinomas and 0 of 21 lung squamous cell carcinomas). The authors used a combination of molecular and histological assays to find that the UPF1 mutations caused UPF1 mis-splicing, loss of UPF1 protein, and impaired NMD, resulting in stable expression of aberrant mRNAs containing premature termination codons that would normally be degraded by NMD. The recurrent, PASC-specific, and focal nature of the reported UPF1 mutations, together with their dramatic effects on NMD activity, suggested that UPF1 mutations are a key feature of PASC biology.

Three subsequent studies of distinct PASC cohorts, however, did not report somatic mutations in UPF1 (*Fang et al., 2017*; *Hayashi et al., 2020*; *Witkiewicz et al., 2015*). This absence of UPF1 mutations is significantly different from the high rate reported by Liu et al. (0 of 34 total PASC samples from three cohorts [*Fang et al., 2017*; *Hayashi et al., 2020*; *Witkiewicz et al., 2015*] vs. 18 of 23 PASC samples from Liu et al; $p<10^{-8}$ by the two-sided binomial proportion test). Although these other studies relied on whole-exome and/or genome sequencing instead of targeted UPF1 gene sequencing, those technologies yield good coverage of the relevant UPF1 gene regions because the affected introns are very short. Given this discrepancy, we sought to directly assess the functional contribution of UPF1 mutations to PASC using a combination of biological and molecular assays.

## Results

We first tested the role of the reported UPF1 mutations during tumorigenesis in vivo. Liu et al. reported that the majority of UPF1 mutations caused skipping of UPF1 exons 10 and 11, disrupting UPF1's RNA helicase domain that is essential for its NMD activity (*Lee et al., 2015*). We therefore modeled UPF1 mutation-induced exon skipping by designing paired guide RNAs flanking Upf1 exons 10 and 11, such that these exons would be deleted upon Cas9 expression (*Figure 1A*). We

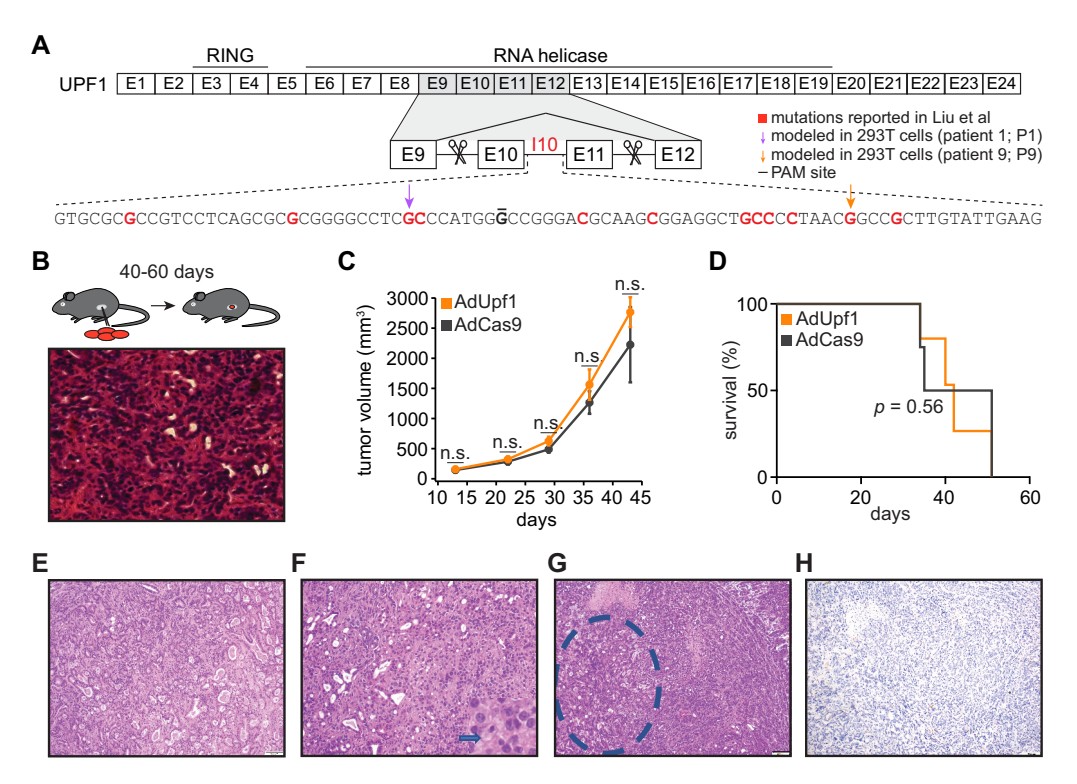

**Figure 1.** *UPF1* mutations do not result in the acquisition of squamous histological features or confer a growth advantage to mutant cells in vivo. (**A**) Schematic of *UPF1* gene structure and corresponding encoded protein domains. Intron 10 (I10) contains the bulk of the mutations reported by Liu et al. Scissors indicate the sites targeted by the paired guide RNAs used to excise exons 10 and 11 (E10 and E11). Red nucleotides represent positions subject to point mutations reported in Liu et al. Arrows indicate specific mutations that we modeled in 293 T cells. The horizontal black line indicates the nucleotide within the protospacer adjacent motif (PAM) site that we mutated to prevent repeated cutting by Cas9 in 293 T cells. (**B**) Top, experimental strategy for testing whether mimicking *UPF1* mis-splicing by deleting exons 10 and 11 promoted pancreatic cancer growth. Mice were orthotopically injected with mouse pancreatic cancer cells (KPC cells: *Kras*^G12D; *Trp53*^R172H/null; Pdx1-Cre) lacking *Upf1* exons 10 and 11. Bottom, hematoxylin and eosin (H and E) stain of pancreatic tumor tissue harvested from the mice. (**C**) Line graph comparing tumor volume between mice injected with control (AdCas9; Cas9 only) or treatment (AdUpf1; Cas9 with *Upf1*-targeting guide RNAs) KPC cells. Tumor volume measured by ultrasound imaging. Error bars, standard deviation computed over surviving animals (n = 10 at first time point). n.s., not significant (p>0.05). p-values at each timepoint were calculated relative to the control group with an unpaired, two-tailed *t*-test. (**D**) Survival curves for the control (AdCas9) or treatment (AdUpf1) cohorts. Error bars, standard deviation computed over biological replicates (n = 10, each group). p-value was calculated relative to the control group by a logrank test. (**E**) Representative hematoxylin and eosin (H and E) staining of a pancreatic tumor resulting from orthotopic injection of control KPC cells displaying features of a moderately to poorly differentiated pancreatic ductal adenocarcinoma. Tumors were composed of medium-size duct-like structures and small tubular glands with lower mucin production. (**F**) Representative H and E image illustrating a moderately to poorly differentiated pancreatic ductal adenocarcinoma resulting from orthotopic injection of *Upf1*-targeted KPC cells. Depicted here is a section of the poorly differentiated component (arrow), which was characterized by solid sheets of tumor cells with large eosinophilic cytoplasms and marked nuclear polymorphism. (**G**) Representative H and E image of a pancreatic tumor resulting from orthotopic injection of *Upf1*-targeted KPC cells. The dashed circle marks a moderately differentiated component; the remainder is poorly differentiated. (**H**) Representative IHC image of a pancreatic tumor resulting from orthotopic injection of *Upf1*-targeted KPC cells for the squamous marker p40 (ΔNp63). No expression of the marker was observed in tumor cells. The online version of this article includes the following source data and figure supplement(s) for figure 1:

**Source data 1.** Source data for mouse tumor volume (*Figure 1C*).
**Source data 2.** Source data for mouse survival (*Figure 1D*).
**Figure supplement 1.** Validation experiments in mouse KPC cells.

chose mouse pancreatic cancer cells (KPC cells: *Kras*^G12D; *Trp53*^R172H/null; Pdx1-Cre) as a model system. KPC cells are defined by mutations affecting KRAS and p53 (encoded by *Kras* and *Trp53* in mouse) that also occur in the vast majority of PASC cases (*Borazanci et al., 2015*; *Fang et al., 2017*; *Hayashi et al., 2020*), making them a genetically appropriate system. We delivered *Upf1*-targeting paired guide RNAs to KPC cells using recombinant adenoviral vectors and confirmed that guide

delivery resulted in the production of *UPF1* mRNA lacking exons 10 and 11 and a corresponding reduction in full-length UPF1 protein levels (*Figure 1—figure supplement 1A–G*). We injected subcloned control and *Upf1*-targeted KPC cells into the tails of the pancreata of B6 albino mice (n = 10 mice per treatment) and monitored tumor growth and animal survival. We detected no significant differences in tumor volume or survival in mice implanted with control or *Upf1*-targeted KPC cells (*Figure 1B–D* and *Figure 1—source datas 1* and *2*). Tumors derived from control as well as *Upf1*-targeted cells displayed similar histopathological features characteristic of moderately to poorly differentiated pancreatic ductal adenocarcinomas (*Figure 1E–G*). Moderately differentiated areas were composed of medium to small duct-like structures or tubules with lower mucin production, while poorly differentiated components were characterized by solid sheets or nests of tumor cells with large eosinophilic cytoplasms and large pleomorphic nuclei. No squamous differentiation was identified by histomorphologic evaluation and no expression of the squamous marker p40 (ΔNp63) was detected (*Figure 1H* and *Supplementary file 1a*). We concluded that inducing the reported *Upf1* exon skipping in vivo had no detectable effects on pancreatic cancer growth or acquisition of adenosquamous features in the KPC model. However, there are several important caveats to our data. First, we cannot rule out the possibility that inducing *Upf1* exon skipping in a different model system or cell type could influence tumorigenesis. Second, as our assays were performed in the complex setting of in vivo tumorigenesis, we cannot infer how inducing the reported *Upf1* exon skipping might affect tumor cell proliferation in the controlled setting of in vitro growth. Third, as accurate measurement of *Upf1* splicing and UPF1 protein isoforms was only possible for KPC cells prior to orthotopic injection, we cannot infer how the relative frequencies of mis-spliced *UPF1* mRNA and the resulting truncated proteins may have changed during tumorigenesis.

We next assessed molecular phenotypes induced with *UPF1* mutations. Liu et al. measured the effects of each mutation on *UPF1* splicing using a minigene assay, in which each mutation was introduced into a plasmid containing a small fragment of the *UPF1* gene that was subsequently transfected into 293 T cells. Liu et al. concluded that all reported *UPF1* mutations caused dramatic *UPF1* mis-splicing that disrupted key protein domains that are essential for UPF1 function in NMD. Minigenes are common tools for studying splicing, but they are frequently spliced less efficiently than endogenous genes, presumably because they are gene fragments that lack potentially important sequence features that promote splicing and incompletely capture the close relationship between chromatin and splicing (*Luco et al., 2011*; *Naftelberg et al., 2015*).

We modeled *UPF1* mutations in 293 T cells in order to mimic Liu et al.'s experimental strategy, but introduced mutations into their endogenous genomic contexts rather than using minigenes. We selected two distinct *UPF1* mutations in intron 10 for these studies. We selected IVS10+31G>A (patient 1; P1) because it was reportedly recurrent across three different patients (making it equally or more common than any other mutation) and induced strong mis-splicing on its own (36% mis-spliced mRNA, versus 0% for wild-type *UPF1*); we selected IVS10-17G>A (patient 9; P9) because it had one of the strongest effects on splicing (90% mis-spliced mRNA). IVS10+31G>A was present in a homozygous state in two of the three patients carrying it, while IVS10-17G>A was present in a heterozygous state.

We introduced each mutation into its endogenous context by transiently transfecting a plasmid expressing Cas9 and a single guide RNA (sgRNA) targeting *UPF1* intron 10 as well as appropriate donor DNA for homology-directed repair, screened the resulting cells for the desired genotypes, and established clonal lines. The resulting cell lines contained the desired mutations in the correct copy numbers as well as a point mutation disrupting the protospacer adjacent motif (PAM) site (*Figure 2—figure supplement 1A–C*). As neither the PAM site itself nor nearby positions were reported as mutated in Liu et al., we additionally established a cell line in which only the PAM site was mutated as a wild-type control.

We systematically tested the functional consequences of *UPF1* mutations for NMD efficiency, UPF1 protein levels, and *UPF1* splicing. We measured NMD efficiency in our engineered cells using the well-established beta-globin reporter system, which permits controlled measurement of the relative levels of mRNAs that do or do not contain an NMD-inducing premature termination codon, but which are otherwise identical (*Zhang et al., 1998*). We did not observe decreased NMD efficiency in *UPF1*-mutant versus wild-type cells; instead, *UPF1*-mutant cells exhibited evidence of modestly more efficient NMD, although these differences were not statistically significant (*Figure 2A* and *Figure 2—source data 1*). To confirm these results from reporter experiments, we then queried levels of

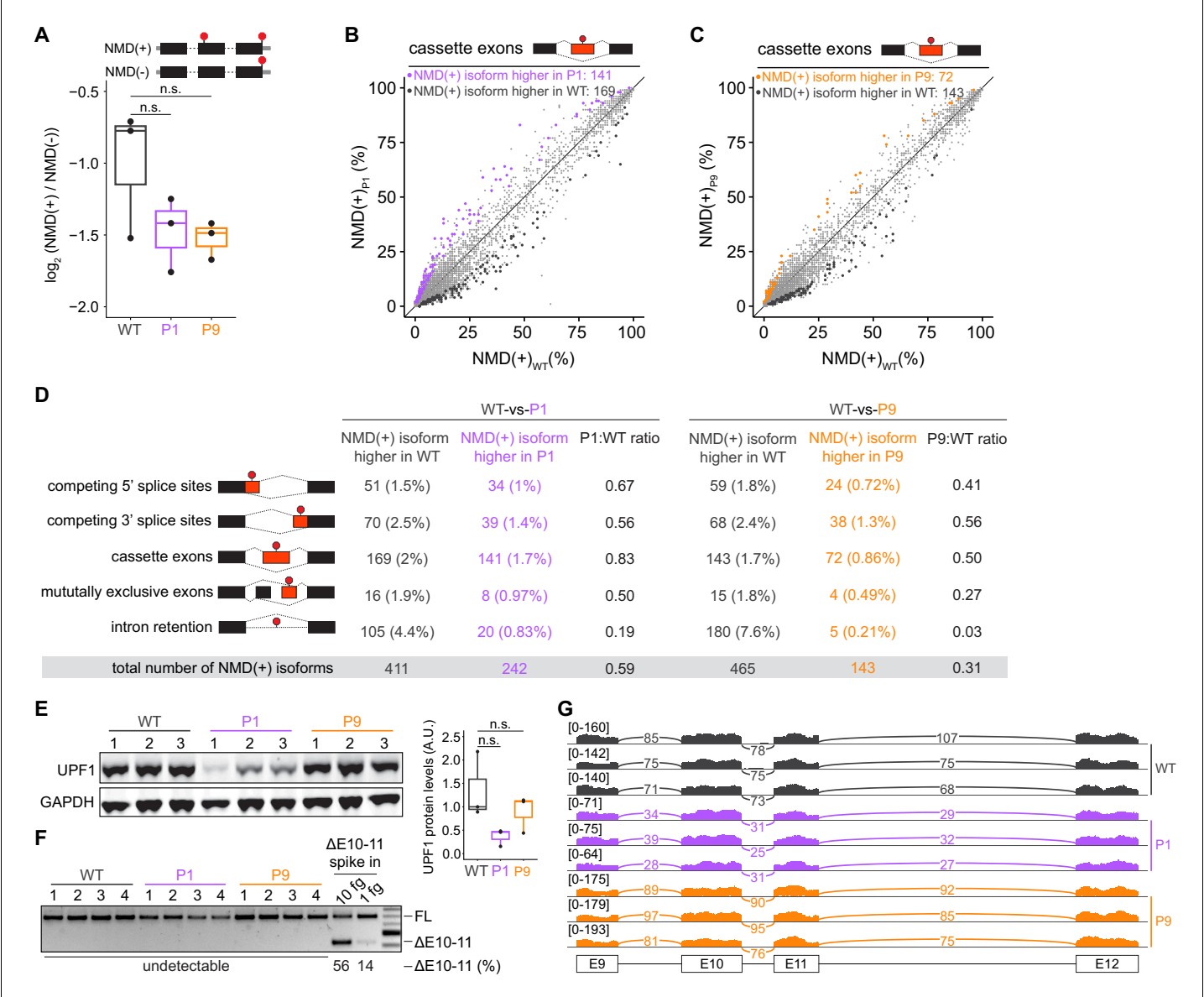

**Figure 2.** Mutations in *UPF1* intron 10 do not inhibit nonsense-mediated mRNA decay (NMD) or cause exon skipping. (**A**) Box plot of NMD efficiency in 293 T cells engineered to contain wild-type (WT) or mutant (P1, P9) *UPF1*. P1 and P9 correspond to the IVS10+31G>A and IVS10-17G>A mutations reported by Liu et al. All cells have the protospacer adjacent motif (PAM) site mutation illustrated in *Figure 1A*. NMD efficiency estimated via the beta-globin reporter assay[11]. Middle line, notches, and whiskers indicate median, first and third quartiles, and range of data. Each point corresponds to a single biological replicate. n.s., not significant (p>0.05). p-values were calculated for each variant relative to the control by a two-sided Mann–Whitney *U* test (p=0.40 for P1, 0.30 for P9). (**B**) Scatter plot showing transcriptome-wide quantification of transcripts containing NMD-promoting features in 293 T cells carrying the *UPF1* mutation that was reportedly observed in patient 1 relative to control, wild-type cells. Each point corresponds to a single isoform that is a predicted NMD substrate (NMD(+)). Purple points represent NMD substrates that are significantly increased in *UPF1*-mutant cells relative to wild-type cells; black points represent NMD substrates that exhibit the opposite behavior. Plot is restricted to NMD substrates arising from differential inclusion of cassette exons. Significantly increased/decreased NMD substrates were defined as transcripts that displayed either an absolute increase/decrease in isoform ratio of ≥10% or an absolute log fold-change in expression of ≥2 with associated p≤0.05 (two-sided *t*-test). (**C**) As (**B**), but for 293 T cells carrying the *UPF1* mutation that was reportedly observed in patient 9. Gold points represent NMD substrates that are significantly increased in *UPF1*-mutant cells relative to wild-type cells. (**D**) Summary of the numbers of NMD substrates arising from differential alternative splicing that exhibit significantly higher or lower levels in *UPF1*-mutant cells relative to wild-type cells. Analysis is identical to (**B**) and (**C**), but extended to the illustrated different types of alternative splicing. (**E**) Left, immunoblot of full-length UPF1 protein for the 293 T cell lines. Each lane represents a single biological replicate with the indicated genotype. GAPDH serves as a loading control. Equal amounts of protein were loaded in each lane (measured by fluorescence). Right, box plot illustrating UPF1 protein levels relative to GAPDH for each genotype. Middle line, notches, and whiskers indicate median, first and third quartiles, and range of data. Each point corresponds to a single biological replicate. Data was quantified with Fiji (v2.0.0). A.U., arbitrary

*Figure 2 continued on next page*

*Figure 2 continued*

units. n.s., not significant (p>0.05). p-values were calculated for each variant relative to the control by a two-sided Mann–Whitney *U* test (p=0.10 for P1, 1.0 for P9). (F) PCR using primers that amplify both full-length *UPF1* mRNA (FL) and mRNA lacking exons 10 and 11 (ΔE10-11). *UPF1* mRNA lacking exons 10 and 11 was only detected in the positive control lanes (ΔE10-11 spike in), in which DNA corresponding to *UPF1* cDNA lacking exons 10 and 11 was synthesized and added to cDNA libraries created from WT cells prior to PCR. Numbers above each lane indicate biological replicates. Numbers below each lane represent the abundance of the lower band as a percentage of total intensity (see Materials and methods). Data was quantified with Fiji (v2.0.0). (G) RNA-seq read coverage across the genomic locus containing *UPF1* exons 9–12 in the indicated 293 T cell lines. Each sample corresponds to a distinct biological replicate. Numbers represent read counts that supported each indicated splice junction (*Katz et al., 2015*).

The online version of this article includes the following source data and figure supplement(s) for figure 2:

**Source data 1.** Source data for qPCR in HEK 293 T cell lines (*Figure 2A*).
**Source data 2.** Source data for western blot in HEK 293 T cell lines (*Figure 2E*).
**Source data 3.** Source data for RT-PCR in HEK 293 T cell lines (*Figure 2F*).
**Figure supplement 1.** Validation experiments in human 293 T cells and raw gel images for western blot and RT-PCR.

endogenous NMD substrates across the transcriptome. We performed high-coverage RNA-seq on each of the three 293 T cell lines that we engineered to lack or contain defined *UPF1* mutations in biological triplicate, quantified transcript expression, and identified differentially expressed transcripts. We focused on NMD substrates arising from alternative splicing, as these are abundant and sensitive biomarkers of NMD efficiency that are internally controlled for gene expression variation (*Feng et al., 2015*). These analyses revealed that neither *UPF1*-mutant cell line exhibited global increases in the expression of NMD substrates relative to wild-type cells. Instead, both *UPF1*-mutant cell lines exhibited modestly lower global levels of endogenous NMD substrates than did wild-type cells, mimicking the trend observed with our NMD reporter experiments. Together, these data confirm that the tested mutations in *UPF1* intron 10 do not affect NMD activity (*Figure 2B–D* and *Supplementary file 1b and c*).

Consistent with similar NMD activity independent of *UPF1* mutational status, *UPF1* mutations did not cause loss of full-length UPF1 protein (*Figure 2E*, *Figure 2—figure supplement 1D*, and *Figure 2—source data 2*). Although UPF1 protein levels varied between the individual cell lines, this variation in UPF1 protein levels was not associated with variation in NMD efficiency and did not segregate with *UPF1* mutational status. We therefore measured the levels of normally spliced and mis-spliced *UPF1* mRNA. We readily detected normally spliced *UPF1* mRNA in all samples by RT-PCR, but found no evidence of mis-spliced *UPF1* mRNA, except in positive control samples in which we spiked in synthesized DNA corresponding to the exon skipping isoform reported in Liu et al. (*Figure 2F*, *Figure 2—figure supplement 1E*, and *Figure 2—source data 3*). We confirmed these results with our RNA-seq data by mapping all reads against all possible splice junctions connecting exons 9, 10, 11, and 12. These analyses revealed no evidence of splice junctions consistent with the reported exon 10 and 11 skipping or other abnormal exon skipping isoforms (*Figure 2G*).

Given the differences between Liu et al.'s findings of common *UPF1* mutations and their absence from subsequent studies of PASC, we wondered whether some of the *UPF1* mutations reported by Liu et al. might correspond to inherited genetic variation rather than somatically acquired mutations. We searched for each mutation reported by Liu et al. within databases compiled by the 1000 Genomes Project, NHLBI Exome Sequencing Project, Exome Aggregation Consortium (ExAC), and the genome aggregation database (gnomAD) (*Auton et al., 2015*; *Karczewski et al., 2020*; *Exome Aggregation Consortium et al., 2016*; *Server EV, 2016*). These databases were constructed from a mix of whole-genome and whole-exome sequencing, both of which are effective for discovering variants within the relevant regions of *UPF1* (because *UPF1* introns 10, 21, and 22 are very short, they are well covered by exon-capture technologies). We found genetic variants identical to 45% (18 of 40) of the reported *UPF1* mutations, one of which is present in the reference human genome. Eighty-nine percent (16 of 18) of *UPF1*-mutant patients had one or more reported mutations that corresponded to standing genetic variation (*Figure 3A–F* and *Supplementary file 1d*). The distribution of overlaps between reported *UPF1* mutations and standing genetic variation depended strongly upon genic context. A large fraction of reported intronic *UPF1* mutations were identical to standing genetic variation, while the majority of reported exonic *UPF1* mutations were not (*Supplementary file 1d*).

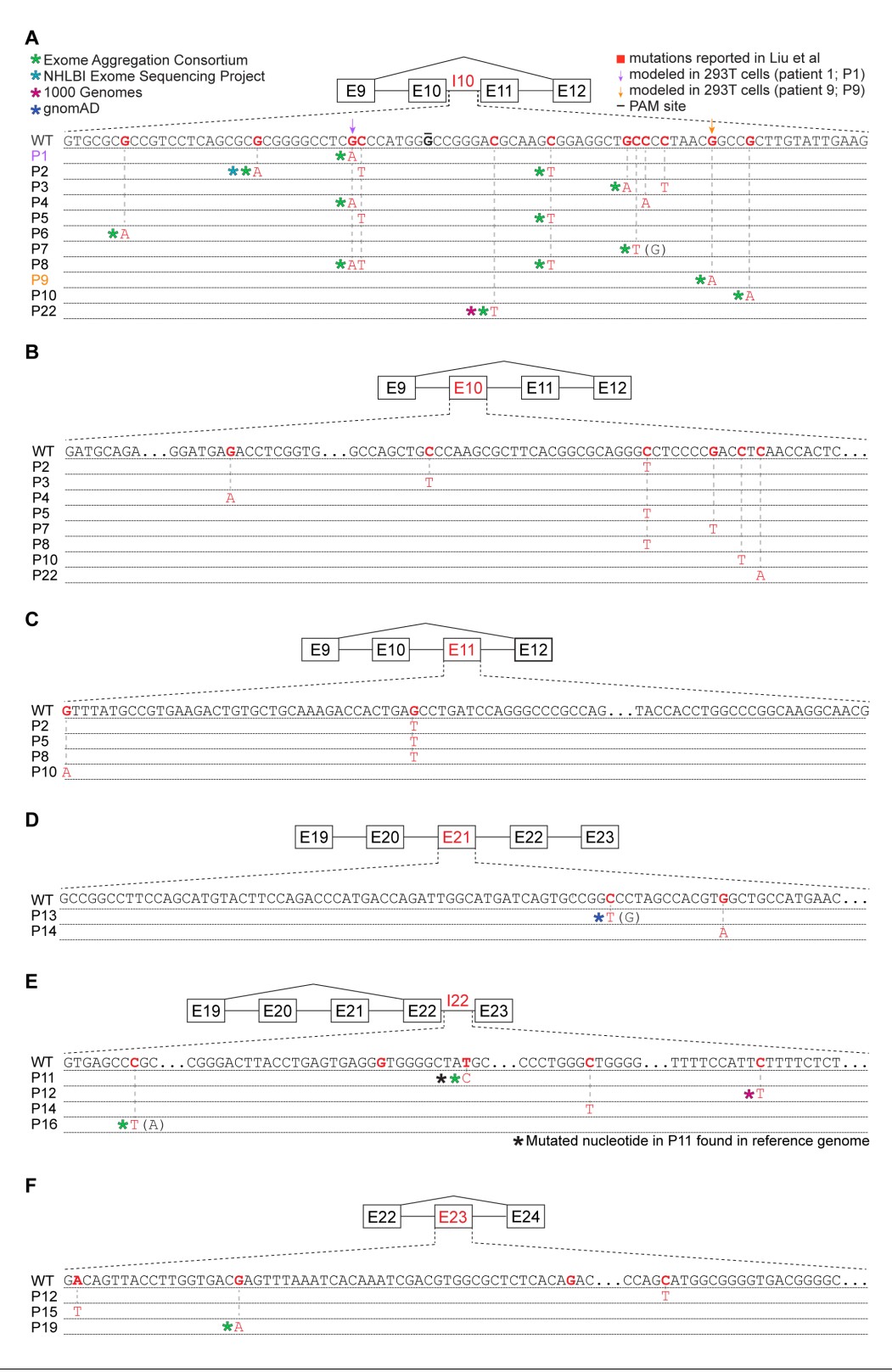

**Figure 3.** Many reported *UPF1* mutations are identical to genetic variants. (**A**) Illustration of the mutations in *UPF1* intron 10 (I10) reported by Liu et al. Each row indicates the wild-type (WT) sequence from the reference human genome or mutations reported by Liu et al. (P1, patient 1). Purple and gold arrows indicate the mutations that we modeled with genome engineering in 293 T cells for patient 1 and patient 9, respectively. Red nucleotides represent positions subject to point mutations reported in Liu et al. The horizontal black line indicates the nucleotide within the protospacer adjacent

*Figure 3 continued on next page*

*Figure 3 continued*

motif (PAM) site that we mutated to prevent repeated cutting by Cas9. Parentheses indicate where we found genetic variation at a reported mutation position that differed from the specific mutated nucleotide reported by Liu et al. (**B**) As (**A**), but for *UPF1* exon 10 (E10). (**C**) As (**A**), but for *UPF1* exon 11 (E11). (**D**) As (**A**), but for *UPF1* exon 21 (E21). (**E**) As (**A**), but for *UPF1* intron 22 (I22). (**F**) As (**A**), but for *UPF1* exon 23 (E23).

Our discovery that a large fraction of the reported *UPF1* mutations are present in databases of germline genetic variation was surprising for two reasons. First, when strongly cancer-linked mutations occur as germline variants, they frequently manifest as cancer predisposition syndromes. However, no such relationship is known for *UPF1* genetic variants, despite their reportedly high prevalence as identical somatic mutations in PASC. Second, *UPF1* is essential for embryonic viability and development in mammals (*Medghalchi et al., 2001*), zebrafish (*Wittkopp et al., 2009*), and Drosophila (*Avery et al., 2011*). As Liu et al. reported that all *UPF1* mutations caused mis-splicing that is expected to disable UPF1 protein function (*Liu et al., 2014*), then those mutations should be incompatible with life when present as inherited genetic variants. Our finding that two reported mutations had no effect on *UPF1* splicing when introduced into their endogenous genomic contexts offers a way to explain this incongruity, at least for the two reported lesions that we studied.

Given these discrepancies, we next sought to verify the somatic nature of the *UPF1* mutations described in Liu et al., which was reportedly determined by sequencing both tumors and patient-matched controls. The GenBank accession codes reported in Liu et al. corresponded to short nucleotide sequences containing *UPF1* mutations, without corresponding data for patient-matched controls. We contacted the senior author (Dr. YanJun Lu) to request primary sequencing data from patient-matched tumor and normal samples, but neither primary sequencing data from matched samples nor the samples themselves were available.

To further explore whether *UPF1* is recurrently mutated in PASC, we reanalyzed sequencing data from *Fang et al., 2017* to manually search for *UPF1* mutations (*Supplementary file 1e-f*). We focused on the two loci that contained all mutations reported by Liu et al. (*UPF1* exons 10-11 and exons 21–23). Because the relevant introns are very short, they were well covered by both the whole-exome and whole-genome sequencing used by Fang et al. Using relaxed mutation-calling criteria to maximize sensitivity (details in Materials and methods), we identified somatic *UPF1* mutations in samples from 6 of 17 PASC patients. However, those mutations exhibited genetic characteristics expected of passenger, not driver, mutations. None of those *UPF1* mutations matched the *UPF1* mutations reported by Liu et al., and only one was present at an allelic frequency equal to the allelic frequency of mutant *KRAS*, which is a known driver and which we detected in samples from all PASC patients (median allelic frequencies of 12% versus 34% for *UPF1* versus *KRAS* mutations). Furthermore, we also identified *UPF1* mutations in samples from patients with non-adenosquamous tumors (3 of 34 pancreatic ductal adenocarcinomas), whereas Liu et al. reported finding no *UPF1* mutations in non-adenosquamous pancreatic cancers (0 of 29). In concert with the reports of *Witkiewicz et al., 2015* and *Hayashi et al., 2020* of finding no *UPF1* mutations in their PASC samples, these analyses suggest that *UPF1* is not a frequent or adenosquamous-specific mutational target in most PASC cohorts.

## Discussion

*UPF1*'s role in the pathogenesis of PASC has been unclear and controversial given the seeming discrepancies between its mutational spectrum in different PASC cohorts. Although it is difficult to conclusively prove that a specific genetic change does not promote cancer, we were unable to detect biological or molecular changes arising from two mutations reported by Liu et al. *UPF1*'s status as an essential gene and our discovery that many reported *UPF1* mutations occur as germline genetic variants of no known pathogenicity together suggest that other *UPF1* mutations reported by Liu et al. could similarly represent genetic differences that do not functionally contribute to PASC. Our study highlights the need for continued study of the PASC mutational spectrum in order to understand the molecular basis of this disease.

## Materials and methods

### Construction of mouse KPC cells carrying a deletion of *Upf1* exons 10 and 11

Mouse KPC cells ($Kras^{G12D}$; $Trp53^{R172H/null}$; Pdx1-Cre) were obtained from Dr. Robert Vonderheide and were cultured in DMEM (GIBCO) supplemented with 10% fetal bovine serum (FBS) and 1% Penicillin/Streptomycin (GIBCO). All cell lines were incubated at 37°C and 5% $CO_2$. Guide RNAs targeting mouse *Upf1* introns 9 and 11 were cloned into a paired guide expression vector (px333) as previously described (*Maddalo et al., 2014*). An EcoRI-XhoI fragment containing the double U6-sgRNA cassette and Flag-tagged Cas9 was then ligated into the EcoRI-XhoI-digested pacAd5 shuttle vector. Recombinant adenoviruses were generated by Viraquest (Ad-Upf1 and Ad-Cas9) or purchased from the University of Iowa (Ad-Cre). KPC cells were infected with ($5 \times 10^6$ PFU) of Ad-Cas9 or Ad-Upf1 in each well of a 6-well plate.

Genomic DNA was extracted 48 hr post infection to confirm excision of *Upf1* exons 10 and 11. For PCR analysis of genomic DNA, cells were collected in lysis buffer (100 nM Tris-HCl at pH 8.5, 5 mM EDTA, 0.2% SDS, 200 mM NaCl supplemented with fresh proteinase K at a final concentration of 100 ng/mL). Genomic DNA was extracted with phenol–chloroform–isoamylic alcohol and precipitated in ethanol, and the DNA pellet was dried and resuspended in double-distilled water. For RT-PCR, total RNA was extracted with TRIzol (Life Technologies) following the manufacturer's instructions. cDNA was synthesized using SuperScript III (ThermoFisher) following the manufacturer's instructions.

### Immunoblots in mouse KPC cells

Cells were lysed in 1X RIPA buffer with protease and phosphatase inhibitors. Fifteen micrograms of protein was separated on 4–10% acrylamide/bisacrylamide gels, transferred onto PVDF membranes, and blocked for 1 hr in 5% milk in $1\times$ TBST. The membranes were incubated with rabbit UPF1 antibody (CST #9435) used at 1:1000 dilution in 5% BSA $1\times$ TBST overnight at 4°C or mouse Tubulin (Sigma T9206) used at 1:2000 dilution in 5% milk $1\times$ TBST for 1 hr at room temperature. Following primary antibody incubation, the membranes were washed three times with $1\times$ TBST buffer at room temperature and probed with rabbit or mouse horseradish peroxidase-linked secondary antibody (1:5000; ECL NA931 mouse, NA934V Rabbit). The western blot signal was detected using the ECL Prime (RPN322) kit and the blot was exposed to an X-Ray film, which was developed using the Konica Minolta SRX 101A film processor.

### Tumorigenicity and metastasis assays

KPC cells carrying *Upf1* ΔE10-11 or an empty Cas9 control were mixed in 1:1 Matrigel (BD Biosciences) and simple media to a final concentration of 100,000 cells in 30 μL of total volume. Cells were orthotopically implanted into the tails of the pancreata of B6 albino mice (Charles River). Ten mice were implanted with each stable, genetically engineered cell line. Tumor growth was measured weekly via 3D-ultrasound starting at 10 days post-implantation. For survival assessment, animals were sacrificed following the endpoints approved by IACUC: (i) animals showing signs of significant discomfort, (ii) ascites or overt signs of tumor metastasis or gastrointestinal bleeding (blood in stool), (iii) animals losing >15% of their body weight, and (iv) animals with tumors >2 cm in diameter. Investigators responsible for monitoring and measuring the xenografts of individual tumors were not blinded. All animal studies were performed in accordance with institutional and national animal regulations. Animal protocols were approved by the Memorial Sloan-Kettering Cancer Center Institutional Animal Care and Use Committee (14-08-009 and 11-12-029). Power analysis was used to determine appropriate sample size to detect significant changes in animal median survival, which was based on previous survival analyses (*Escobar-Hoyos et al., 2020*). Survival curves and statistics were performed using PRISM.

### Immunohistochemistry (IHC) and histopathological analysis

Paraffin sections were dewaxed in xylene and hydrated in graded alcohols. Endogenous peroxidase activity was blocked by immersing the slides in 1% hydrogen peroxide in PBS for 15 min. Pretreatment was performed in a steamer using 10 mM citrate buffer (pH 6.0) for 30 min. Sections were

incubated overnight with a primary rabbit polyclonal antibody against p40-DeltaNp63 (Abcam, ab166857) diluted at a ratio of 1:100. Sections were washed with PBS and incubated with an appropriate secondary antibody followed by avidin–biotin complexes (Vector Laboratories, Burlingame, CA, PK-6100). The antibody reaction was visualized with 3–3' diaminobenzidine (Sigma, D8001) followed by counterstaining with hematoxylin. Tissue sections were dehydrated in graded alcohols, cleared in xylene, and mounted. For p40 (ΔNp63) IHC, expression was defined based on nuclear labeling.

## Culture and genome engineering of HEK 293 T cells

HEK 293 T cells were cultured in DMEM media (GIBCO) supplemented with 10% FBS (GIBCO), 100 IU penicillin, and 100 mg/mL streptomycin (PenStrep, GIBCO). Cells were cultured at 37°C and 5% $CO_2$. Cells were split at a ratio of 1:10 once they reached 90–100% confluency as needed. Cell lines were authenticated using ATCC fingerprinting and tested regularly for mycoplasma contamination.

Guide RNAs for all cell lines were designed using the GuideScan 1.0 software package (*Perez et al., 2017*) and sequences were chosen among those predicted to have the highest cutting efficiency and specificity scores. DNA oligos for all guide RNAs were synthesized by IDT and amplified using primers that appended homology arms to facilitate ligation into the pX459/Cas9 expression plasmid (*Ran et al., 2013*) by Gibson assembly (*Gibson et al., 2009*). Gibson assembly reactions were transformed into NEB Stable Competent *E. coli* and resulting sgRNA expression plasmids were amplified and purified using standard protocols.

Ultramers for homology directed repair (HDR) were designed using previously described strategies (*Richardson et al., 2016*) and synthesized by IDT, Inc. HEK 293 T cells were transiently transfected with 1 µg/mL sgRNA/pX459 Cas9 expression plasmid and 20 nM HDR Ultramer using Lipofectamine 2000 that was diluted in Opti-MEM Reduced Serum Medium (ThermoFisher). Cells were incubated at 37°C for 24 hr, at which time the transfection medium was replaced with DMEM media (GIBCO) supplemented with 10% FBS (GIBCO), 100 IU penicillin, 100 mg/mL streptomycin (PenStrep, GIBCO), and 2 mg/mL puromycin. Cells were then incubated for 48–72 hr in DMEM media (GIBCO) supplemented with 10% FBS (GIBCO), 100 IU penicillin, 100 mg/mL streptomycin (PenStrep, GIBCO), and 2 mg/mL puromycin, at which time genomic DNA was extracted and regions of interest were amplified using the appropriate oligos.

Genome engineering was validated using genomic DNA as follows. Amplicons from genomic DNA PCR were ligated into vectors using the Zero Blunt TOPO PCR cloning system (ThermoFisher) and the presence of the desired mutations was validated using Sanger sequencing (GENEWIZ). Polyclonal cell populations were then diluted and sorted into 96-well plates using a BD FACS Aria II flow cytometer (BD Biosciences), such that each well contained on average one cell, which were grown in DMEM media (GIBCO) supplemented with 20% FBS (GIBCO), 100 IU penicillin, and 100 mg/mL streptomycin (PenStrep, GIBCO). Once cells in 96-well plates reached confluency, they were transferred to 24-well plates and allowed to grow to confluency for genomic DNA extraction and Sanger sequencing.

## NMD efficiency measurement

NMD efficiency was estimated using the beta-globin reporter system (*Zhang et al., 1998*). HEK 293 T cells engineered with the reported mutations were plated at 10–15% confluency on pol-L-lysine coated 12-well plates. Cells were co-transfected 24 hr later with 1 µg of phCMV-MUP plasmid (transfection control) and 1 µg of either pmCMV-Gl-Norm (normal termination codon) or pmCMV-Gl-39Ter (premature termination codon) using Lipofectamine 3000 (Invitrogen) according to the manufacturer's protocol. After 48 hr the cells were close to confluency, at which time they were lysed using 1 mL of Trizol Reagent (Invitrogen) per well. The lysate was collected, and total RNA was extracted according to the manufacturer's protocol. The RNA was further purified and DNase treated using the Direct-zol RNA MiniPrep Kit (Zymo Research) according to the manufacturer's protocol.

Residual plasmid DNA was removed using DNaseI (Amplification Grade, Invitrogen) according to the manufacturer's protocol from 600 ng of the extracted RNA. cDNA synthesis was then performed using SuperScript IV Reverse Transcriptase (Invitrogen) with oligo dT primers according to the manufacturer's protocol. The cDNA synthesis reaction was diluted 1:50 and 4 µL was used for a 10 µL

qPCR reaction with PowerUp SYBR Green Master Mix (ThermoFisher) and primers specific for the reporter mRNA diluted to a working concentration of 100 nM for each primer. qPCR reactions were performed in technical triplicate for three different biological replicates in 384-well plates (Thermo-Fisher) using an ABI QuantStudio 5 Real-Time PCR System (ThermoFisher). The levels of pmCMV-Gl-Norm and pmCMV-Gl-39Ter cDNA were normalized to phCMV-MUP mRNA abundance for each sample, and levels of pmCMV-Gl-39Ter cDNA were plotted relative to levels of pmCMV-Gl-Norm for each replicate in each cell type. Statistical analysis was performed using PRISM.

## Immunoblots in HEK 293 T cells

Cells were lysed using a buffer containing 150 mM NaCl, 1% NP-40, and 50 mM Tris pH 8.0 supplemented with a phosphatase inhibitor (ThermoFisher) and protease inhibitor (ThermoFisher). After the lysis buffer was added, cells were frozen at −80°C and thawed for three cycles, and then incubated on ice for 15 min. The lysed cells were centrifuged at $10,000 \times g$ for 15 min, and the supernatant was collected to determine total protein concentration. Total protein concentration was determined using a Qubit (ThermoFisher) and 20 µg of total protein was used for electrophoresis. Following electrophoresis, protein was transferred to nitrocellulose membrane (Novex) in transfer buffer containing 10% methanol overnight at 4°C. The blot was blocked with Odyssey Blocking Buffer (LI-COR Biosciences) for 1 hr at room temperature and probed using a 1:1000 dilution of 0.514 mg/mL UPF1 antibody (Abcam, 109363) overnight with shaking at 4°C. Following overnight incubation, the blot was washed three times with 1× TBST buffer at room temperature and probed with rabbit secondary antibody (IRDye 680RD goat anti-rabbit) for 1 hr at room temperature. The blot was then imaged and UPF1 abundance was quantified using band intensity in Fiji (v2.0.0). Statistical analysis was performed using PRISM.

## Isoform detection by RT-PCR in HEK 293 T cell lines

Total RNA was extracted using the RNeasy Plus Mini Kit (Qiagen). cDNA was synthesized using oligo dT primers and SuperScript IV Reverse Transcriptase (ThermoFisher) following the manufacturer's protocol. PCR was carried out with primers targeting *UPF1* exons 9 and 12 using Q5 High-Fidelity DNA Polymerase (NEB). PCR products were run in a 2% agarose slab gel and stained with ethidium bromide for visualization by UV shadowing (Bio-Rad Molecular Imager Gel Doc XR+).

A positive control for the amplification of the truncated *UPF1* variant missing exons 10 and 11 was synthesized as a double-stranded DNA gBlock (IDT). Two different amounts (10 fg and 1 fg) of this 'spike in' control was added to separate PCRs containing cDNA from wild-type HEK 293 T cells and amplified in the same manner as described above.

To quantify the degree of exon skipping (percent ΔE10-11), a background subtraction across the entire gel was first performed in Fiji using a rolling ball radius of 100 pixels. Next, the integrated density of each band was determined, and the density of the lower band in each lane was divided by the total density in that same lane by summing the integrated densities of the upper and lower bands.

## Reanalysis of genomic DNA sequencing data from Fang et al.

Whole-genome and whole-exome sequencing data from *Fang et al., 2017* were downloaded from the Sequence Read Archive (accession number SRP107982) and mapped to the *UPF1* (chr19:18940305–18979266; hg19/GRCh37 assembly) and *KRAS* (chr12:25356390–25405419; hg19/GRCh37 assembly) gene loci. The first 10 nt of all reads were trimmed off due to low sequencing quality and the trimmed reads were mapped with Bowtie v1.0.0 (*Langmead et al., 2009*) with the arguments '-v 3 k 1 m 1 –best –strata –minins 0 –maxins 1000 –fr'. Mapped reads were visualized in IGV (*Robinson et al., 2011*). Mutation/genetic variant calling thresholds (read coverage depth ≥9 reads and ≥2 reads supporting the mutation/variant) were chosen in order to allow detection of a hotspot *KRAS* mutation (G12 or G13) in every PASC sample in the cohort. That criteria ensured that our thresholds were appropriate for discovering known cancer driver mutations in all samples. A genetic difference from the reference genome was defined as a somatic mutation if it was called in a tumor sample but not in the corresponding patient-matched normal control sample.

## Genome annotations

Genome and transcriptome annotations for mapping RNA-seq data to the human (NCBI GRCh37/ UCSC hg19) genome were generated as described previously (*Dvinge et al., 2014*). Briefly, transcriptome annotations from Ensembl (*Flicek et al., 2013*) were merged with isoform annotations from the MISO v2.0 database (*Katz et al., 2010*) and the UCSC knownGene track (*Meyer et al., 2013*). NMD substrates were defined as those isoforms containing a premature termination codon >50 nt upstream of the last exon–exon junction.

## RNA-seq read mapping

RNA-seq reads were mapped to the transcriptome with RSEM v1.2.4 (*Li and Dewey, 2011*) and Bowtie v1.0.0 (*Langmead et al., 2009*), where RSEM was modified to invoke Bowtie with the '-v2' option. Reads that are unaligned after this transcriptome mapping were then aligned to the genome with TopHat v2.0.8 (*Trapnell et al., 2009*), as well as mapped to a database of splice junctions that was defined by creating all possible co-linear combinations of 5′ and 3′ splice sites within every gene. A final file of aligned reads was created by merging the read alignments from TopHat with the read alignments from RSEM.

## Differential splicing analysis

MISO v2.0 (*Katz et al., 2010*) was used to quantify isoform expression for alternative splicing events. Differentially spliced events were defined as those that met the following criteria: (1) had at least 20 informative reads (reads that uniquely distinguish between isoforms of a given splicing event), (2) exhibited either an absolute change in isoform ratio $\geq$10% or an absolute fold-change $\geq$2, and (3) had an associated $p \leq 0.05$ (computed using a two-sided t-test). Differential splicing analyses were restricted to splicing events arising from U2-type (major) introns, which constitute >99% of all introns, in order to ensure that no potential confounding effects arose from intron type. Splicing events were defined as NMD relevant if at least one, but not all, of the child isoforms was predicted NMD substrates.

## Acknowledgements

We thank the members of the Bradley, Ventura, and Leach laboratories for comments and suggestions. We specifically thank the following individuals for their technical help and support: Olivera Grbovic-Huezo for pancreatic injections, Paul Ogrodowski and Jonathan Bermeo for assistance with mouse work and tissue harvest, Maria S Jiao and the MSK Center For Comparative Medicine and Pathology Facility for p40 IHC, and Miles Wilkinson for discussing our findings. JTP was supported in part by the NIH/NCI (T32 CA009657). DU was supported in part by the NIH/NIGMS (T32 GM007270). RK was supported in part by the NIH/NCI (T32 CA160001). SDL was supported in part by the NIH/NCI (R01 CA204228). AV was supported in part by the Cycle for Survival's Equinox Innovation Award in Rare Cancers and a Functional Genomics Initiative grant (AV). RKB is a Scholar of The Leukemia and Lymphoma Society (1344–18).

## Additional information

### Funding

| Funder | Grant reference number | Author |
|---|---|---|
| National Cancer Institute | T32 CA009657 | Jacob T Polaski |
| National Institute of General Medical Sciences | T32 GM007270 | Dylan B Udy |
| National Cancer Institute | T32 CA160001 | Ram Kannan |
| National Cancer Institute | R01 CA204228 | Steven D Leach |
| Leukemia and Lymphoma Society | 1344-18 | Robert K Bradley |

The funders had no role in study design, data collection and interpretation, or the decision to submit the work for publication.

## Author contributions
Jacob T Polaski, Ram Kannan, Conceptualization, Investigation, Writing - original draft; Dylan B Udy, Luisa F Escobar-Hoyos, Gokce Askan, Investigation; Steven D Leach, Supervision; Andrea Ventura, Conceptualization, Supervision; Robert K Bradley, Conceptualization, Supervision, Writing - original draft

## Author ORCIDs
Jacob T Polaski https://orcid.org/0000-0001-6570-1789
Robert K Bradley https://orcid.org/0000-0002-8046-1063

## Ethics
Animal experimentation: All animal studies were performed in accordance with institutional and national animal regulations. Animal protocols were approved by the Memorial Sloan-Kettering Cancer Center Institutional Animal Care and Use Committee (14-08-009 and 11-12-029).

## Decision letter and Author response
Decision letter https://doi.org/10.7554/eLife.62209.sa1
Author response https://doi.org/10.7554/eLife.62209.sa2

# Additional files
## Supplementary files
• Supplementary file 1. (**a**) Histopathological classification of pancreatic tumors derived from orthotopic injection of control or *Upf1*-targeted KPC cells. Tumors displaying >25% squamous differentiation were classified as positive for squamous features. (**b**) Differentially spliced isoforms identified in 293 T cells engineered to have the 'patient 1' *UPF1* genotype relative to WT control cells. (**c**) Differentially spliced isoforms identified in 293 T cells engineered to have the 'patient 9' *UPF1* genotype relative to WT control cells. (**d**) Overlap between *UPF1* mutations reported by Liu et al. and genetic variants present in the 1000 Genomes, NHLBI Exome Sequencing Project, ExAC, and gnomAD databases. Genomic coordinates are specified with respect to the GRCh37/hg19 genome assembly. (**e**) Summary statistics of somatic mutations in *UPF1* identified in our reanalysis of whole-exome and whole-genome sequencing data from Fang et al. (**f**) All genetic differences in *UPF1* and *KRAS* from the reference genome, including both somatic mutations and inherited genetic variants, that we identified in our reanalysis of all samples from Fang et al.'s cohorts.

• Transparent reporting form

## Data availability
RNA-seq data generated as part of this study have been deposited in the Gene Expression Omnibus (accession number GSE163517). All original gel images are provided.

The following dataset was generated:

| Author(s) | Year | Dataset title | Dataset URL | Database and Identifier |
|---|---|---|---|---|
| Polaski JT, Udy DB, Escobar-Hoyos LF, Askan G, Leach SD, Ventura A, Kannan R, Bradley RK | 2020 | The origins and consequences of UPF1 variants in pancreatic adenosquamous carcinoma | https://www.ncbi.nlm.nih.gov/geo/query/acc.cgi?acc=GSE163517 | NCBI Gene Expression Omnibus, GSE163517 |

The following previously published dataset was used:

| Author(s) | Year | Dataset title | Dataset URL | Database and Identifier |
|---|---|---|---|---|
| Fang Y, Su Z, Xie J, Xue R, Qi Ma, Li Y, Zhao Y, Song Z, Lu X, Li H, Peng C, Bai F, Shen B | 2017 | pancreatic ductal adenocarcinoma and panreatic adenosquamous carcinoma sequencing | https://trace.ncbi.nlm.nih.gov/Traces/sra/?study=SRP107982 | NCBI Sequence Read Archive, SRP107982 |

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

# Appendix 1

**Appendix 1—key resources table**

| Reagent type (species) or resource | Designation | Source or reference | Identifiers | Additional information |
|---|---|---|---|---|
| Gene (*H. sapiens*) | *UPF1* | Ensembl | ENSG00000005007 | |
| Gene (*M. musculus*) | *Upf1* | Ensembl | ENSMUSG00000058301.8 | |
| Strain, strain background (*M. musculus*) | Mouse | NCI Charles River | Charles River: C57BL/6 albino | Mice for pancreatic injections |
| Strain, strain background (*E. coli*) | NEB Stable Competent *E. coli* (High efficiency) | New England Biolabs | C3040 | Chemically competent |
| Genetic reagent (*M. musculus*) | AdUpf1 | This paper | N/A | Adenovirus expressing CRISPR gRNAs targeting mouse *Upf1* introns 9-11 |
| Cell line (*H. sapiens*) | HEK 293T | ATCC | CRL-11268 | Human cell line to model patient mutations |
| Cell line (*M. musculus*) | KPC | Generated from the PDX-1-Cre; LSL-*Kras*^G12D/+; LSL *Trp53*^R172H/+ | N/A | Murine cell line to model *Upf1* exon skipping mutations (*Hingorani et al., 2005*) Provided by Robert Vonderheide |
| Antibody | Anti-human UPF1 (rabbit monoclonal) | Abcam | Cat No. ab10936 | WB: (1:1000) |
| Antibody | Anti-human GAPDH (rabbit polyclonal) | Abcam | Cat No. ab9485 | WB: (1:1000) |
| Antibody | Anti-rabbit secondary antibody (goat monoclonal) | Abcam | Cat No. ab216777 | WB: (1:10000) IRDye 680RD |
| Antibody | Anti-mouse UPF1 (rabbit monoclonal) | Cell signaling technology | Cat No.9435 | WB: (1:1000) |
| Antibody | Anti-mouse tubulin (mouse monoclonal) | Sigma-Aldrich | Cat No. T9206 | WB: (1:2000) |
| Antibody | Anti-rabbit secondary antibody (from donkey) | Amersham | NA93V | WB: (1:5000) |
| Antibody | Anti-mouse secondary antibody (from sheep) | Amersham | NA931 | WB: (1:5000) |

*Continued on next page*

*Appendix 1—key resources table continued*

| Reagent type (species) or resource | Designation | Source or reference | Identifiers | Additional information |
|---|---|---|---|---|
| Antibody | Anti-mouse p40-ΔNp63 (rabbit polyclonal) | Abcam | Cat No. ab166857 | WB: (1:100) |
| Recombinant DNA reagent | pX459/Cas9 expression plasmid | Addgene | Cat No. 48139 | *Ran et al., 2013* |
| Recombinant DNA reagent | phCMV-MUP | PMID:9671053 | Plasmid | Control for transfection efficiency of pmCMV-GI-Norm and pmCMV-GI-39Ter |
| Recombinant DNA reagent | pmCMV-GI-Norm | PMID:9671053 | Plasmid | Transient transfection construct coding for full-length β-globin |
| Recombinant DNA reagent | pmCMV-GI-39Ter | PMID:9671053 | Plasmid | Transient transfection construct coding for truncated β-globin with PTC at amino acid 39 |
| Sequence-based reagent | mUpf1_F | This paper | PCR primer | GGTGATGAGATTGCTATTGAGC |
| Sequence-based reagent | mUpf1_R | This paper | PCR primer | TGTTCCTGATCTGGTTGTGC |
| Sequence-based reagent | mUpf1-intron_9-gDNA_F | This paper | Guide DNA Oligo | CACCGTTGTGAGGGCCATACCCTTG |
| Sequence-based reagent | mUpf1-intron_9-gDNA_R | This paper | Guide DNA Oligo | AAACCAAGGGTATGGCCCTCACAAC |
| Sequence-based reagent | mUpf1-intron_11-gDNA_F | This paper | Guide DNA Oligo | CACCGCCGTTGAGCTGATGGTGGCT |
| Sequence-based reagent | mUpf1-intron_11-gDNA_R | This paper | Guide DNA Oligo | AAACAGCCACCATCAGCTCAACGGC |
| Sequence-based reagent | hUPF1_F | This paper | Genomic DNA PCR primer | AAAACGTTTGCCGTGGATGAG |
| Sequence-based reagent | hUPF1_R | This paper | Genomic DNA PCR primer | CACATAGAGAGCGGTAGGCA |
| Sequence-based reagent | hUPF1-gDNA_F | This paper | Guide DNA oligo | GCGCGCGGGGCCTCGCCCAT |
| Sequence-based reagent | hUPF1-patient_1-HDR_R | This paper | DNA HDR ultramer | GCTCAGTGGTCTTTGCAGCACAGTCTTCACGGCATAAACCTTCAATACAAGCGGCCGTTAGGGGCAGCCTCCGCTTGCGTCCCGGGCCATGGGTGAGGCCCCGCGCGCTGAGGACGGCGCGCACCTG |

*Continued on next page*

*Appendix 1—key resources table continued*

| Reagent type (species) or resource | Designation | Source or reference | Identifiers | Additional information |
|---|---|---|---|---|
| Sequence-based reagent | hUPF1-patient_9-HDR_R | This paper | DNA HDR ultramer | GCTCAGTGGTCTTTGCAGC ACAGTCTTCACGGCATAA ACCTTCAATACAAGCGGCT GTTAGGGGCAGCCTCCGCT TGCGTCCCGGGCCATGGG CGAGGCCCCGCGCGCTGA GGACGGCGCGCACCTG |
| Sequence-based reagent | hUPF1-PAM-control-HDR_R (wild type) | This paper | DNA HDR ultramer | GCTCAGTGGTCTTTGCAG CACAGTCTTCACGGCATA AACCTTCAATACAAGCGGC CGTTAGGGGCAGCCTCCG CTTGCGTCCCGGGCCATG GGCGAGGCCCCGCGCGC TGAGGACGGCGCGCACCTG |
| Sequence-based reagent | hUPF1-RT-PCR_F | This paper | RT-PCR primer | GGATGAGATATGCCTGCGGT |
| Sequence-based reagent | hUPF1-RT-PCR_R | This paper | RT-PCR primer | TTCTCGTCGGCAGACGACAG |
| Sequence-based reagent | Positive control gBlock to detect *UPF1* splice variant (ΔE10-11) | This paper | DNA gBlock | ACATGCGGCTCATGCAGG GGGATGAGATATGCCTGCG GTACAAAGGGGACCTTGCG CCCCTGTGGAAAGGGATCG GCCACGTCATCAAGGTCCC TGATAATTATGGCGATGAGA TCGCCATTGAGCTGCGGAG CAGCGTGGGTGCACCTGTG GAGGTGACTCACAACTTCC AGGTGGATTTTGTGTGGAA GTCGACCTCCTTTGACAGG CCGGTGCTGGTGTGTGCTC CGAGCAACATCGCCGTGGA CCAGCTAACGGAGAAGATCC ACCAGACGGGGCTAAAGGT CGTGCGCCTCTGCGCCAAGA GCCGTGAGGCCATCGACTCC CCGGTGTCTTTTCTGGCCCT GCACAACCAGATCAGGAACA TGGACAGCATGCCTGAGCTG CAGAAGCTGCAGCAGCTGAA AGACGAGACTGGGGAGCTG TCGTCTGCCGACGAGAAGC GGTACCGGGCCTTGAAGC GCACCGCAGAGAGAG AGCTGCTGATG |
| Sequence-based reagent | mMup1-qPCR_F | PMID:25564732 | qPCR primer | GACCTATCCAATGCC AATCG (exon 5/6 junction) |
| Sequence-based reagent | mMup1-qPCR_R | PMID:25564732 | qPCR primer | GATGATGGTGGAG TCCTGGT (exon 7) |
| Sequence-based reagent | hβ-globin-qPCR_F | PMID:25564732 | qPCR primer | GCTCGGTGCCTTT AGTGATG (exon 2) |
| Sequence-based reagent | mβ-globin-qPCR_R | PMID:25564732 | qPCR primer | CCCAGCACAAT CACGATCATA (exon 3, mouse specific) |

*Continued on next page*

*Appendix 1—key resources table continued*

| Reagent type (species) or resource | Designation | Source or reference | Identifiers | Additional information |
|---|---|---|---|---|
| Commercial assay or kit | RNeasy Plus Mini Kit | Qiagen | Cat No. 79654 | |
| Commercial assay or kit | Zero Blunt TOPO PCR cloning system | ThermoFisher | Cat No. K280020 | |
| Chemical compound, drug | Phosphatase inhibitor | ThermoFisher | Cat No. A32959 | |
| Chemical compound, drug | Protease inhibitor | Thermofisher | Cat No. A32963 | |
| Chemical compound, drug | Penicillin/ Streptomycin | GIBCO | Cat No. 15070063 | |
| Chemical compound, drug | Lipofectamine 2000 | ThermoFisher | Cat No. 11668030 | |
| Chemical compound, drug | Lipofectamine 3000 | Invitrogen | Cat No. L3000001 | |
| Chemical compound, drug | Puromycin | ThermoFisher | Cat No. A1113803 | |
| Software, algorithm | GuideScan v1.0 | PMID:28263296 | http://www.guidescan.com/ | |
| Software, algorithm | Fiji v2.0.0 | ImageJ | https://imagej.net/Fiji | |
| Software, algorithm | RSEM v1.2.4 | PMID:21816040 | deweylab.github.io/RSEM/RRID:SCR_013027 | |
| Software, algorithm | Bowtie v1.0.0 | PMID:19261174 | github.com/BenLangmead/bowtie/; RRID:SCR_005476 | |
| Software, algorithm | TopHat v2.0.8b | PMID:19289445 | ccb.jhu.edu/software/tophat/index.shtml RRID:SCR_013035 | |
| Software, algorithm | MISO v2.0 | PMID:21057496 | genes.mit.edu/burgelab/miso/ RRID:SCR_003124 | |
| Software, algorithm | IGV v2.3.90 | Thorvaldsdottir | software.broadinstitute.org/software/igv/ RRID:SCR_011793 | |
| Software, algorithm | Prism v7.0 | GraphPad Prism v7.0 | http://www.graphpad.com/ RRID:SCR_002798 | |
| Other | SuperScript IV Reverse Transcriptase | ThermoFisher | Cat No. 18090010 | |
| Other | Q5 High-Fidelity DNA Polymerase | New England Biolabs | Cat No. NEB #M0491 | |
| Other | PowerUp SYBER Green Master Mix | ThermoFisher | Cat No. A25742 | |
| Other | RNase-Free DNase Set | Qiagen | Cat No. 79254 | |

