## [Decision Letter]

**Acceptance summary:**

The authors have sought to address what has become a considerably debated topic of whether mutations in *Upf1* are tumorigenic in pancreatic adenosquamous carcinoma. Specifically, the authors introduced *Upf1* mutants found in pancreatic tumors into pancreatic adenosquamous carcinoma cells, and found they did not provide significant advantage for tumor progression. Moreover, the authors described how a significant percentage of *Upf1* mutants observed in pancreatic carcinoma are also present as germline variants in the human population, raising further doubts about their potential role as cancer drivers. Altogether, this work provides further evidence as to whether *Upf1* disruptive mutations represent driving factors in pancreatic adenosquamous carcinoma.

**Decision letter after peer review:**

Thank you for submitting your article "The origins and consequences of *UPF1* variants in pancreatic adenosquamous carcinoma" for consideration by *eLife*. Your article has been reviewed by three peer reviewers, and the evaluation has been overseen by a Reviewing Editor and Erica Golemis as the Senior Editor. The following individual involved in review of your submission has agreed to reveal their identity: Wei Li (Reviewer #3).

The reviewers have discussed the reviews with one another and the Reviewing Editor has drafted this decision to help you prepare a revised submission.

Summary:

The authors have sought to address what has become a considerably debated topic of whether mutations in *Upf1* are tumorigenic in pancreatic adenosquamous carcinoma. Specifically, the authors introduced *Upf1* mutants found in pancreatic tumors into pancreatic adenosquamous carcinoma cells, and found they did not provide significant advantage for tumor progression. Moreover, the authors described how a significant percentage of *Upf1* mutants observed in pancreatic carcinoma are also present as variants in the human population, raising further doubts about their potential role as cancer drivers. Altogether, this work provides further evidence as to whether *Upf1* disruptive mutations represent driving factors in pancreatic adenosquamous carcinoma.

Essential Revisions:

Although we append the specific reviewer comments below to aid in your revision, discussion amongst reviewers has generated the following Essential Revision list:

1) Test additional NMD targets: reviewer 1 mentions doing this in the CRISPR-E10/E11 KPC cells and mining RNA-seq data from previous *Upf1* variant data; reviewer 2 mentions analyzing other NMD targets in 293T *Upf1* splicing mutant cells, to extend results beyond use of a reporter assay.

2) Perform in vitro growth competition assay to rigorously determine if CRISPR-Exon10/Exon11 KPC cells display any difference from control KPC cells.

3) Conduct additional database queries regarding *Upf1* mutations and conduct some additional analyses of their data (as described by reviewer 3).

Reviewer #1:

This manuscript identifies that the *UPF1* variants previously reported as frequent somatic mutations in pancreatic adenosquamous carcinoma are actually germline genetic variants with no clear effects on *UPF1* splicing, protein splicing, or nonsense mediated decay. Given that the manuscript challenges a striking finding from a prior study that has not been validated in subsequent studies, it is important to publish to correct the literature. At the same time, several points should be clarified to make sure the data are as comprehensive as possible:

1) In the experiments evaluating the effect of skipping exons 10-11 of *UPF1*, It is surprising that this genetic perturbation in *UPF1* is actually tolerated in these cells as *UPF1* is an essential gene in most cancer cell lines (this point also has likely motivated this current study). Also, the Western blots for *UPF1* protein are not particularly clear (Figure 1—figure supplement 1C) and the fact that the cells don't perturb the growth of KPC cells does not prove that *UPF1* alterations is not tumorigenic. Have the authors checked to see if *UPF1* is downregulated and mis-spliced in the cells following in vivo growth? A simple in vitro competition assay between *UPF1* exon 10-11 targeted cells and control sgRNA cells would also be helpful. It would also be helpful to evaluate if NMD is altered in these cells given these issues.

2) Although it is clear that the authors have used similar minigene assays as were used in the original publication, a more systematic evaluation for potential alteration in NMD with *UPF1* variants (via RNA-seq) would be helpful given that this work questions the prior publication.

3) Do the authors believe that the *UPF1* variants reported as mutations initially in PASC are actually SNPs? The terminology describing what these variants are could be a little clearer in the Abstract and Discussion.

Reviewer #2:

This paper aims to resolve the disparity between one report (Liu et al., 2014), which described somatic mutations in pancreatic adenosquamous carcinoma (PASC) that did not typify normal pancreatic tissue of the patients, and other reports (Witkiewicz et al., 2015; Fang et al., 2017; Hayashi et al., 2020), which did not find these mutations. The authors show here that many (40%) of the mutations described by Liu et al. typify genetic variations in the human population at large, and they suggest that these mutations are not pathogenic, e.g. are not drivers of PASC, and also not somatic but, rather, are genetic in origin.

The authors use CRISPR-Cas9 to generate in mouse pancreatic cancer (KPC) cells, which harbor Kras and Tp53 gene mutations (as do PASC patients), a *Upf1* gene, and thus its product mRNA, lacking exons 10 and 11. Liu et al. previously reported this *Upf1* variant not only inhibits NMD by disrupting *UPF1* helicase activity, but also promotes tumorigenesis. After injection into mice, the authors found no detectable effects on pancreatic cancer growth compared to the injection of control cells.

The authors acknowledge that mice may differ from humans. Thus next, rather than using mini-*UPF1* genes, as did Liu et al., the authors introduced two of the Liu et al. mutations separately into the *UPF1* gene of HEK293T cells. In contrast to Liu et al., the authors found modestly increased NMD efficiency and no evidence of *UPF1* pre-mRNA mis-splicing. The authors note that this makes sense since these mutations are found in people not as somatic mutations but genetic mutations, and thus would not be expected to inhibit NMD given the importance of NMD to aspects of human development in utero and beyond.

This is a very well-written paper describing carefully executed experiments that lead the reader to discount three claims made about *UPF1* gene mutations in PANC as described by Liu et al., namely, that these mutations: (i) have a somatic origin, (ii) lead to *UPF1* pre-mRNA mis-splicing so as to inhibit NMD, and (iii) promote tumorigenesis. The authors are careful not to over-interpret their data.

Results, in reference to Figure 1F. It is unexpected that the variations in *UPF1* protein levels were "uncorrelated with NMD efficiency". Possibly, this reviewer doesn't understand what the authors mean. Please clarify.

Additionally, in this regard, it is better to draw conclusions about NMD efficiency by measuring more than just the efficiency with which mRNA from a reporter construct is targeted for NMD. It is recommended that the authors assay the levels of a few (e.g. three) cellular NMD targets, normalized to the level of their pre-mRNA to control for any changes to gene transcription.

Reviewer #3:

In the manuscript, Polaski et al. compared the reported *UPF1* mutations with a collection of three databases and found 42.5% of these mutations are identical to germline genetic variation. However, most of these overlapped mutations are located within introns, and only present in Exome Aggregation Consortium (ExAC) database (Figure 2). This raised some concerns since the ExAC database mainly report exon variants rather than intron variants. The authors need provide other information such as allele frequency to examine whether these intronic mutations are rare or low-frequency variants.

Another suggestion is that the authors may cross-reference *UPF1* mutations with the recent gnomAD v3 database (Nature 2020), which provided non-coding genetic variants within much better resolution. In addition, most of the other *UPF1* exon mutations are indeed novel as they are not present in any databases (Figure 2—figure supplement 1). The authors need to provide some additional analysis such as separating these two types of variants (exon/intron variants) and analyzing the frequency of overlapped *UPF1* mutations.

---

## [Author Response]

Essential Revisions:Although we append the specific reviewer comments below to aid in your revision, discussion amongst reviewers has generated the following Essential Revision list:1) Test additional NMD targets: reviewer 1 mentions doing this in the CRISPR-E10/E11 KPC cells and mining RNA-seq data from previous Upf1 variant data; reviewer 2 mentions analyzing other NMD targets in 293T Upf1 splicing mutant cells, to extend results beyond use of a reporter assay.

We thank reviewers 1 and 2 for these excellent suggestions to confirm that levels of endogenous substrates are not affected by the *UPF1* mutations reported by Liu et al. We addressed this point by performing high-coverage RNA-seq in biological triplicate on wild-type 293T cells as well as the two distinct *UPF1*-mutant 293T cell lines which we generated (mimicking the mutations reported in “patient 1” and “patient 9” in Liu et al.). We used these RNA-seq data to quantify levels of thousands of NMD substrates arising from alternative splicing across the transcriptome an approach which we have previously used to study NMD perturbation (e.g., Feng et al., 2015) – and identify NMD substrates which were significantly differentially expressed between wild-type and *UPF1*-mutant cell lines. These analyses confirmed the results of our NMD reporter experiments, and clearly demonstrated that levels of endogenous substrates are not higher in either *UPF1*-mutant cell line relative to wild-type cells. The revised text describing these new experiments reads:

“To confirm these results from reporter experiments, we then queried levels of endogenous NMD substrates across the transcriptome. […] Together, these data confirm that the tested mutations in *UPF1* intron 10 do not affect NMD activity (Figure 2B-D and Figure 2—source data 1-2).”

Figure 2B illustrates the abundance of NMD substrates arising from differential cassette exon inclusion (“poison exons,” which we have previously shown to be sensitive biomarkers of NMD activity; see Feng et al., 2015 and Thomas et al., Nature Genetics, 2020) in wild-type (WT; x axis) and *UPF1*-mutant (P1; y axis) engineered 293T cells. Global levels of these NMD substrates are similar between the two genotypes.

We additionally used our new RNA-seq data to confirm our RT-PCR-based conclusion that the two *UPF1* mutations which we studied did not cause exon 10 and 11 skipping. The relevant new text reads:

“We confirmed these results with our RNA-seq data by mapping all reads against all possible splice junctions connecting exons 9, 10, 11, and 12. These analyses revealed no evidence of splice junctions consistent with the reported exon 10 and 11 skipping or other abnormal exon skipping isoforms (Figure 2G and Figure 2—figure supplement 1E).”

Finally, please note that we addressed this reviewer request by studying our *UPF1*-mutant 293T cells, rather than the KPC system, because our 293T cells contain the exact mutations reported by Liu et al. Our KPC cells, in contrast, were genetically engineered to lack exons 10 and 11 of *UPF1* in order to force them to produce the UPF1 mRNA isoforms that Liu et al. reported resulted from *UPF1* mutations based on their minigene assays. However, our studies showed that the reported *UPF1* mutations do not cause production of such aberrant UPF1 mRNA isoforms when the mutations are introduced into their endogenous genomic contexts (rather than in Liu et al.’s minigene reporter).

2) Perform in vitro growth competition assay to rigorously determine if CRISPR-Exon10/Exon11 KPC cells display any difference from control KPC cells.

We thank the reviewer for raising this point. We designed our KPC studies to assess whether production of the reported mis-spliced *UPF1* mRNA resulted in overt differences in tumorigenesis in vivo, including bulk tumor growth, host survival, and acquisition of squamous features. We feel that this was a reasonable approach to test the possible role of *UPF1* mutations in PASC, but agree that our studies do not test how deletion of *UPF1* exons 10 and 11 affects tumor cell growth in vitro.

As our subsequent molecular studies demonstrated that the reported *UPF1* mutations do not induce exon 10 and 11 skipping, we respectfully request that we be permitted to address this point by editing the text to carefully limit our claims, rather than conducting additional experiments. The revised text reads:

“We concluded that inducing the reported *Upf1* exon skipping in vivo had no detectable effects on pancreatic cancer growth or acquisition of adenosquamous features in the KPC model. […] Third, as accurate measurement of *Upf1* spicing and UPF1 protein isoforms was only possible for KPC cells prior to orthotopic injection, we cannot infer how the relative frequencies of mis-spliced UPF1 mRNA and the resulting truncated proteins may have changed during tumorigenesis.”

3) Conduct additional database queries regarding Upf1 mutations and conduct some additional analyses of their data (as described by reviewer 3).

We thank reviewer #3 for this suggestion, as we agree that it will help to clarify this dataset for the reader. We addressed this point by:

– Cross-referencing every variant in the supplementary file with the gnomAD v3 database. We confirmed that all previously called variants are also present in gnomAD v3.

– Adding a column to the supplementary file that lists the allele frequency of each variant (as reported in gnomAD v3).

– Separating exonic and intronic variants.

We additionally edited the text in appropriate places to note, for example, that a large fraction of reported intronic mutations were identical to inherited variants, while many reported exonic mutations were not.

Itemized responses to specific reviewer comments follow below.

Reviewer #1:This manuscript identifies that the UPF1 variants previously reported as frequent somatic mutations in pancreatic adenosquamous carcinoma are actually germline genetic variants with no clear effects on UPF1 splicing, protein splicing, or nonsense mediated decay. Given that the manuscript challenges a striking finding from a prior study that has not been validated in subsequent studies, it is important to publish to correct the literature. At the same time, several points should be clarified to make sure the data are as comprehensive as possible:1) In the experiments evaluating the effect of skipping exons 10-11 of UPF1, It is surprising that this genetic perturbation in UPF1 is actually tolerated in these cells as UPF1 is an essential gene in most cancer cell lines (this point also has likely motivated this current study).

As the reviewer points out, the fact that *UPF1* is an essential gene in most cells partially motivated our study, as that fact was difficult to reconcile with the dramatic *UPF1* mis-splicing evident in Liu et al.’s reporter assay. We introduced the CRISPR/Cas9 construct that deleted exons 10 and 11 of *Upf1* in KPC cells via an adenoviral vector, which induced gDNA deletion as desired; however, this deletion was incomplete (as is typical for adenoviral delivery of CRISPR/Cas9 constructs). This is evident in Figure 1—figure supplement 1A.

We hypothesize that most cells likely did not have disruption of both alleles of *Upf1*. This is consistent with prior reports that although deletion of both *Upf1* alleles is embryonic lethal in mice, mice that are heterozygous for *Upf1* deletion are phenotypically normal (Medghalchi et al., 2001).

Also, the Western blots for UPF1 protein are not particularly clear (Figure 1—figure supplement 1C) and the fact that the cells don't perturb the growth of KPC cells does not prove that UPF1 alterations is not tumorigenic. Have the authors checked to see if UPF1 is downregulated and mis-spliced in the cells following in vivo growth?

We thank the reviewer for this point. We confirmed that *Upf1* was mis-spliced and produced abnormal, short protein isoforms by RT-PCR and Western blot after introducing our CRISPR/Cas9 construct in vitro (in the figure noted by the reviewer). We performed these assays on pure KPC cells prior to orthotopic injection in order to generate the most controlled data. We agree that it would be very interesting to study *Upf1* splicing and protein isoforms following in vivo growth; however, as the resulting tumors are a heterogenous population of KPC cells and host cells, the resulting data would be difficult to interpret. We note this point in the revised text as follows:

“We concluded that inducing the reported *Upf1* exon skipping in vivo had no detectable effects on pancreatic cancer growth or acquisition of adenosquamous features in the KPC model. […] Third, as accurate measurement of *Upf1* spicing and *UPF1* protein isoforms was only possible for KPC cells prior to orthotopic injection, we cannot infer how the relative frequencies of mis-spliced *UPF1* mRNA and the resulting truncated proteins may have changed during tumorigenesis.”

A simple in vitro competition assay between UPF1 exon 10-11 targeted cells and control sgRNA cells would also be helpful. It would also be helpful to evaluate if NMD is altered in these cells given these issues.

We thank the reviewer for raising this point. We designed our KPC studies to assess whether production of the reported mis-spliced *UPF1* mRNA resulted in overt differences in tumorigenesis in vivo, including bulk tumor growth, host survival, and acquisition of squamous features. We feel that this was a reasonable approach to test the possible role of *UPF1* mutations in PASC, but agree that our studies do not test how deletion of *UPF1* exons 10 and 11 affects tumor cell growth in vitro.

As our subsequent molecular studies demonstrated that the reported *UPF1* mutations do not induce exon 10 and 11 skipping, we respectfully request that we be permitted to address this point by editing the text to carefully limit our claims, rather than conducting additional experiments. The revised text reads:

“We concluded that inducing the reported *Upf1* exon skipping in vivo had no detectable effects on pancreatic cancer growth or acquisition of adenosquamous features in the KPC model. […] Third, as accurate measurement of *Upf1* spicing and UPF1 protein isoforms was only possible for KPC cells prior to orthotopic injection, we cannot infer how the relative frequencies of mis-spliced *UPF1* mRNA and the resulting truncated proteins may have changed during tumorigenesis.”

2) Although it is clear that the authors have used similar minigene assays as were used in the original publication, a more systematic evaluation for potential alteration in NMD with UPF1 variants (via RNA-seq) would be helpful given that this work questions the prior publication.

We thank the reviewer for this excellent suggestion to confirm that levels of endogenous substrates are not affected by the *UPF1* mutations reported by Liu et al. We addressed this point by performing high-coverage RNA-seq in biological triplicate on wild-type 293T cells as well as the two distinct *UPF1*-mutant 293T cell lines which we generated (mimicking the mutations reported in “patient 1” and “patient 9” in Liu et al.). We used these RNA-seq data to quantify levels of thousands of NMD substrates arising from alternative splicing across the transcriptome – an approach which we have previously used to study NMD perturbation (e.g., Feng et al., 2015) – and identify NMD substrates which were significantly differentially expressed between wild-type and *UPF1*-mutant cell lines. These analyses confirmed the results of our NMD reporter experiments, and clearly demonstrated that levels of endogenous substrates are not higher in either *UPF1*-mutant cell line relative to wild-type cells. These new data are illustrated in Figure 2B-D of the revised manuscript.

3) Do the authors believe that the UPF1 variants reported as mutations initially in PASC are actually SNPs? The terminology describing what these variants are could be a little clearer in the Abstract and Discussion.

We thank the reviewer for raising this important point for clarification. Yes, we believe that a substantial fraction of the *UPF1* variants that were initially reported as somatic mutations by Liu et al. are actually SNPs, not somatic mutations. However, we were unable to obtain access to primary sequencing data or primary patient materials from the senior author of Liu et al. in order to test whether the reported mutations were actually SNPs. We therefore are hesitant to directly claim that the reported mutations are truly SNPs, since we weren’t able to formally disprove Liu et al.’s claim that they are somatic mutations using Liu et al.’s primary samples or data. We hope that this is reasonable.

Reviewer #2:This paper aims to resolve the disparity between one report (Liu et al., 2014), which described somatic mutations in pancreatic adenosquamous carcinoma (PASC) that did not typify normal pancreatic tissue of the patients, and other reports (Witkiewicz et al., 2015; Fang et al., 2017; Hayashi et al., 2020), which did not find these mutations. The authors show here that many (40%) of the mutations described by Liu et al. typify genetic variations in the human population at large, and they suggest that these mutations are not pathogenic, e.g. are not drivers of PASC, and also not somatic but, rather, are genetic in origin.The authors use CRISPR-Cas9 to generate in mouse pancreatic cancer (KPC) cells, which harbor Kras and Tp53 gene mutations (as do PASC patients), a Upf1 gene, and thus its product mRNA, lacking exons 10 and 11. Liu et al. previously reported this Upf1 variant not only inhibits NMD by disrupting UPF1 helicase activity, but also promotes tumorigenesis. After injection into mice, the authors found no detectable effects on pancreatic cancer growth compared to the injection of control cells.The authors acknowledge that mice may differ from humans. Thus next, rather than using mini-UPF1 genes, as did Liu et al., the authors introduced two of the Liu et al. mutations separately into the UPF1 gene of HEK293T cells. In contrast to Liu et al., the authors found modestly increased NMD efficiency and no evidence of UPF1 pre-mRNA mis-splicing. The authors note that this makes sense since these mutations are found in people not as somatic mutations but genetic mutations, and thus would not be expected to inhibit NMD given the importance of NMD to aspects of human development in utero and beyond.This is a very well-written paper describing carefully executed experiments that lead the reader to discount three claims made about UPF1 gene mutations in PANC as described by Liu et al., namely, that these mutations: (i) have a somatic origin, (ii) lead to UPF1 pre-mRNA mis-splicing so as to inhibit NMD, and (iii) promote tumorigenesis. The authors are careful not to over-interpret their data.

We appreciate the reviewer’s feedback, and particularly the comment that we are “careful not to over-interpret” our data. Given that our studies unexpectedly did not support the conclusions of Liu et al., we tried to be as cautious and careful as possible.

Results, in reference to Figure 1F. It is unexpected that the variations in UPF1 protein levels were "uncorrelated with NMD efficiency". Possibly, this reviewer doesn't understand what the authors mean. Please clarify.

We agree that it is interesting that we did not observe a relationship between *UPF1* protein levels in each engineered cell line and NMD efficiency in the same cell line. We hypothesize that this discordance may arise, at least in part, by the fact that *UPF1* phosphorylation is important for NMD to proceed, but most *UPF1* protein is hypophosphorylated (Kurosaki et al., Genes and Development, 2014). The modest differences in total (independent of phosphorylation) *UPF1* protein levels that we observed across our engineered cell lines are presumably insufficient to substantially affect NMD efficiency. Although we feel that this hypothesis is plausible, we did not include it in the text as it is speculative. We modestly revised the text to emphasize the key point, which is that the *UPF1*-mutant cell lines did not exhibit dramatically lower UPF1 protein levels (as expected based on the results of Liu et al.):

“Although *UPF1* protein levels varied between the individual cell lines, this variation in *UPF1* protein levels was not associated with variation in NMD efficiency and did not segregate with *UPF1* mutational status.”

Additionally, in this regard, it is better to draw conclusions about NMD efficiency by measuring more than just the efficiency with which mRNA from a reporter construct is targeted for NMD. It is recommended that the authors assay the levels of a few (e.g. three) cellular NMD targets, normalized to the level of their pre-mRNA to control for any changes to gene transcription.

We thank the reviewer for this excellent suggestion, which was also suggested by reviewer 1, to confirm that levels of endogenous substrates are not affected by the *UPF1* mutations reported by Liu et al. We addressed this point by performing high-coverage RNA-seq in biological triplicate on wild-type 293T cells as well as the two distinct *UPF1*-mutant 293T cell lines which we generated (mimicking the mutations reported in “patient 1” and “patient 9” in Liu et al.). We used these RNA-seq data to quantify levels of thousands of NMD substrates arising from alternative splicing across the transcriptome – an approach which we have previously used to study NMD perturbation (e.g., Feng et al., 2015) – and identify NMD substrates which were significantly differentially expressed between wild-type and *UPF1*-mutant cell lines. These analyses confirmed the results of our NMD reporter experiments, and clearly demonstrated that levels of endogenous substrates are not higher in either *UPF1*-mutant cell line relative to wild-type cells. These new data are illustrated in Figure 2B-D of the revised manuscript.

Reviewer #3:In the manuscript, Polaski et al. compared the reported UPF1 mutations with a collection of three databases and found 42.5% of these mutations are identical to germline genetic variation. However, most of these overlapped mutations are located within introns, and only present in Exome Aggregation Consortium (ExAC) database (Figure 2). This raised some concerns since the ExAC database mainly report exon variants rather than intron variants. The authors need provide other information such as allele frequency to examine whether these intronic mutations are rare or low-frequency variants.

We thank the reviewer for raising this important point. In many ways, we were lucky with our study, as the unusual shortness of the relevant introns (intron 10: 85 bp; intron 21: 110 bp; intron 22: 265 bp) meant that they are well-covered by exome sequencing data. In the revised manuscript, we include a new column for each variant in Supplementary file 4 (the table of reported mutations and their overlap with inherited variants). One reported mutation is extremely common – it is actually present in the reference human genome assembly – while most were relatively rare.

Another suggestion is that the authors may cross-reference UPF1 mutations with the recent gnomAD v3 database (Nature 2020), which provided non-coding genetic variants within much better resolution.

We thank the reviewer for this suggestion. We cross-referenced every reported mutation in Supplementary file 4 against the gnomAD v3 database. We confirmed the presence of all variants that we reported in our initial submission, and additionally found one more variant (rs956365722; exon 21) that is present in gnomAD v3, but not in either the ExAC or 1000 Genomes databases.

In addition, most of the other UPF1 exon mutations are indeed novel as they are not present in any databases (Figure 2—figure supplement 1). The authors need to provide some additional analysis such as separating these two types of variants (exon/intron variants) and analyzing the frequency of overlapped UPF1 mutations.

This is an excellent point. We agree that it is interesting to note that a large fraction of the reported intronic mutations overlap with genetic variants, while many reported exonic mutations do not. In the revised Supplementary file 4, we separate exonic and intronic variants to clarify to the reader which regions contain which genetic variants. We also highlight this point in the revised text as follows:

“The distribution of overlaps between reported *UPF1* mutations and standing genetic variation depended strongly upon genic context. A large fraction of reported intronic *UPF1* mutations were identical to standing genetic variation, while the majority of reported exonic *UPF1* mutations were not (Supplementary file 4).”